# Does Multimodality Help in Deep Learning-Based Structural Heart Disease Detection?

**Young Sang Choi**[1]                                    YOUNG.SANG.CHOI@COLUMBIA.EDU

**Shalmali Joshi**[1,2]                                          SJ3261@CUMC.COLUMBIA.EDU

**Linyuan Jing**[3]                                                    CQR9002@NYP.ORG

**Pierre Elias**[1,3,4]                                        PAE2115@CUMC.COLUMBIA.EDU

[1] *Department of Biomedical Informatics, Columbia University, New York, NY, USA*

[2] *Data Science Institute, Columbia University, New York, NY, USA*

[3] *NewYork-Presbyterian Hospital/Columbia University Irving Medical Center, New York, NY, USA*

[4] *Seymour, Paul, and Gloria Milstein Division of Cardiology, Department of Medicine, Columbia University Irving Medical Center, New York, NY, USA*

**Editors:** Accepted for publication at MIDL 2024

## Abstract

Structural heart disease (SHD) is typically diagnosed using transthoracic echocardiograms (TTEs), a modality underutilized in the United States. We investigate what combination of common clinical modalities in electrocardiograms (ECGs), posteroanterior view chest X-rays, and structured electronic health record (EHR) data can detect SHD labels generated with an TTE unseen by the model. Our experiments show that ECG-based models in both unimodal and multimodal settings performed best and that the inclusion of additional modalities with a late-fusion approach can give a marginal performance improvement.

**Keywords:** multimodal learning, cardiology, structural heart disease, deep learning.

## 1. Introduction

Transthoracic echocardiograms (TTEs) are the modality of choice for diagnosing structural heart disease (SHD) (Members et al., 2021). Despite being low-risk and non-invasive, TTEs are often underutilized in the United States due to diagnostic stewardship and non-clinical factors such as competing financial incentives (Papolos et al., 2016). In contrast, electrocardiograms (ECGs) and chest X-rays (CXRs) are among the most common forms of imaging in hospital settings. Motivated by capturing both electrophysiology through ECGs and anatomical structure via CXRs, we investigate which combination of modalities in standard 12-lead electrocardiograms, posteroanterior view chest radiographs, and tabular data containing subject demographics from electronic health records (EHRs) and clinician interpretations of ECGs can best detect structural heart disease labels generated using an TTE unseen by the model.

## 2. Methods

**Subject population and dataset.** We collected data from 12,587 unique patients who visited the NewYork-Presbyterian hospital system in New York City between 2009 and 2019. Our cohort had a mean age of $62.97 \pm 16.40$ years, 7,075 (56.21%) female subjects,

and 2,295 (18.23%) subjects with a diagnosis of SHD. For each subject, we selected their first TTE date and selected the ECG and CXR collected on the closest date irrespective of modality order. Only subjects with all three exams taken within 365 days of each other were included in our dataset and each subject had only one modality triplet and label. For the tabular data, we collected both demographic (patient sex and age at TTE) and ECG-generated features (atrial rate, ventricular rate, PR interval, QRS duration, corrected QT interval, and QT-QTc ratio). We applied a temporal data split, where subjects with TTEs taken on or after 2018 were allocated to the test set. The split between training and validation sets was done randomly with the remaining subjects using a 80:20 ratio.

**Preprocessing.** For all CXR inputs, we center cropped each image along the short edge to make square dimensions, applied Contrast Limited Adaptive Histogram Equalization (CLAHE) (Pizer et al., 1987) with `clip_limit=0.02`, and resized to $224 \times 224$ pixels. We then converted all images from grayscale to RGB and applied ImageNet-1K (IN-1K) (Deng et al., 2009) normalization. Additionally, all ECGs were collected at or downsampled to 250 Hz for 10 seconds to dims (1, 2500, 12). We normalized the waveforms along ECG channels using the respective means and standard deviations from the training set. For our tabular data, we mean imputed missing values and applied standard normalization.

**Models.** For CXR classification, we used a DenseNet-121 (DN) (Huang et al., 2017) model after performing model selection between DN, ResNet-v2-50 (RN) (He et al., 2016b), and Vision Transformer-Base (ViT) (Dosovitskiy et al., 2021) classifiers with a single node output layer pretrained with IN-1K (DN) or IN-21K (RN, ViT) (Deng et al., 2009) based on availability in timm (Wightman, 2019). For ECG classification, we chose a 1-D variant of ResNet-18 (RN-1D) with standard initialization (He et al., 2016a). Our tabular model is a logistic regression classifier with L2 regularization. For the multimodal setting, models with CXRs and ECGs had DN and RN-1D encoders with output size 512 and an additive fusion layer. If tabular features were used, they were concatenated to the encoder outputs. The embeddings were then passed through a fully connected output layer with a single node.

**Training procedure and model selection.** Table 1 contains selected training hyperparameters by model. All neural networks were trained with binary cross-entropy loss with class weights, batch size of 256 for 30 epochs with an AdamW (Loshchilov and Hutter, 2019) optimizer. For all models, we ran hyperparameter sweeps with initial learning rates (LRs) {0.01,0.003,0.001,0.0003} both with and without a cosine LR scheduler (Loshchilov

Table 1: Model-specific training hyperparameters. Tab: tabular. LR: initial learning rate. Sched: learning rate scheduler. Aug: data augmentation. Parentheses denote 95% confidence intervals.

| Input Modality | Model | LR | Sched | Aug | Val AUROC ↑ |
|---|---|---|---|---|---|
| Tab Only | Logistic Regression | - | - | - | 0.740 (0.701,0.774) |
| CXR Only | DenseNet-121 (DN) | 0.003 | None | Yes | 0.720 (0.681,0.755) |
| ECG Only | ResNet-1D (RN-1D) | 0.001 | Cosine | No | 0.844 (0.817,0.873) |
| CXR+Tab | DN+Tab | 0.003 | None | No | 0.764 (0.728,0.799) |
| ECG+CXR | RN-1D+DN | 0.003 | Cosine | No | 0.848 (0.816,0.875) |
| ECG+Tab | RN-1D+Tab | 0.0003 | Cosine | No | 0.839 (0.808,0.869) |
| All | DN+RN-1D+Tab | 0.001 | Cosine | No | 0.859 (0.830,0.888) |

Table 2: Classification metrics on the test set ($n = 1,611$, $16.88\%$ positive). Parentheses denote $95\%$ confidence intervals. Best values by metric are in **bold**.

| | ACCURACY ↑ | AUROC ↑ | AUPRC ↑ | F1 SCORE ↑ |
|---|---|---|---|---|
| Tab | 0.837 (0.818,0.855) | 0.723 (0.686,0.757) | 0.370 (0.316,0.428) | 0.228 (0.170,0.287) |
| CXR | 0.669 (0.646,0.693) | 0.668 (0.630,0.705) | 0.313 (0.263,0.367) | 0.363 (0.320,0.406) |
| ECG | 0.839 (0.821,0.857) | 0.802 (0.773,0.83) | 0.470 (0.410,0.531) | 0.490 (0.435,0.542) |
| CXR+Tab | 0.601 (0.577,0.626) | 0.739 (0.707,0.773) | 0.388 (0.333,0.444) | 0.394 (0.357,0.431) |
| ECG+CXR | 0.783 (0.762,0.803) | 0.816 (0.787,0.844) | 0.498 (0.429,0.562) | **0.512 (0.467,0.556)** |
| ECG+Tab | **0.842 (0.826,0.859)** | 0.805 (0.778,0.832) | 0.482 (0.421,0.548) | 0.411 (0.351,0.466) |
| All | 0.750 (0.729,0.773) | **0.822 (0.793,0.849)** | **0.513 (0.443,0.576)** | 0.499 (0.455,0.544) |

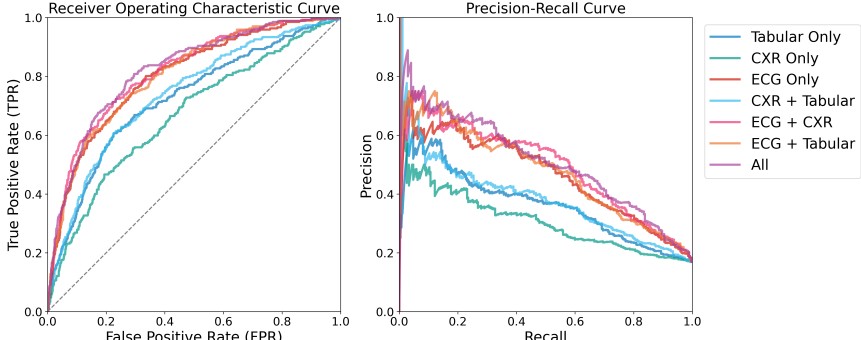

Figure 1: Left: ROC curves for all models for SHD detection. Right: PR curves. Unimodal and multimodal models with ECG inputs outperformed the other models. Multimodal models have a small performance uplift compared to unimodal ones.

and Hutter, 2017). CXR-only classifiers were trained without augmentation, or with horizontal and vertical shifts, random erasing (Zhong et al., 2020), and random crop and resizing augmentations. ECG only RN-1D models were trained without augmentations, or with combinations of time masking and frequency masking. To keep the number of experiments tractable, we did not experiment with augmentations for the multimodal models. We selected the architecture and training procedure with the lowest bootstrapped mean validation AUROC. All experiments were performed using Python 3.12.2, PyTorch 2.2.1 (Paszke et al., 2019), timm 0.9.16 (Wightman, 2019) and standard Python scientific libraries on an Ubuntu 22.04 server equipped with an Intel Xeon E5-2640 CPU, 128 GB of memory, and a NVIDIA GTX Titan X with 12GB VRAM.

**Results.** Table 2 contains classification metrics and Figure 1 contains classification plots. ECG models in both unimodal and multimodal settings outperformed other models. Adding modalities on top of ECGs offer marginal benefit to SHD detection.

## 3. Conclusion

Multimodal models can outperform unimodal ones in deep-learning based detection of echocardiogram diagnosed structural heart disease as long as electrocardiograms are used as one of the input modalities.

## Acknowledgments

The authors sincerely thank the members of the CRADLE Lab at the Columbia University Irving Medical Center in making our work possible.

## Appendix A. Subject Demographics

|  | All | Train | Test |
|---|---|---|---|
| Unique Subjects | 12587 | 10976 | 1611 |
| Male | 5512 (43.79%) | 4805 (43.78%) | 707 (43.89%) |
| Female | 7073 (56.19%) | 6169 (56.2%) | 904 (56.11%) |
| Age | 62.97 ± 16.4 | 62.97 ± 16.36 | 63.01 ± 16.63 |
| SHD Prevalence | 2295 (18.23%) | 2023 (18.43%) | 272 (16.88%) |
| Echocardiogram Year | 2009 - 2019 | 2009 - 2017 | 2018 - 2019 |
| Days b/w TTE and ECG | 23.4 ± 60.85 | 21.68 ± 57.7 | 35.08 ± 78.05 |
| Days b/w TTE and CXR | 53.48 ± 88.34 | 47.97 ± 82.83 | 91.03 ± 112.35 |
| Days b/w ECG and CXR | 49.41 ± 86.71 | 46.0 ± 83.68 | 72.59 ± 102.12 |
| White | 4687 (37.24%) | 4045 (36.85%) | 642 (39.85%) |
| Black or African American | 1992 (15.83%) | 1655 (15.08%) | 337 (20.92%) |
| Asian | 315 (2.5%) | 258 (2.35%) | 57 (3.54%) |
| Declined | 1275 (10.13%) | 1128 (10.28%) | 147 (9.12%) |
| Missing | 2336 (18.56%) | 2233 (20.34%) | 103 (6.39%) |
| Other | 1982 (15.75%) | 1657 (15.1%) | 325 (20.17%) |
| Atrial Rate | 85.76 ± 39.46 | 85.71 ± 39.46 | 86.12 ± 39.47 |
| Ventricular Rate | 80.76 ± 19.27 | 80.68 ± 19.16 | 81.34 ± 19.98 |
| PR Interval | 159.78 ± 31.61 | 160.12 ± 31.57 | 157.45 ± 31.76 |
| QRS Duration | 91.34 ± 19.39 | 91.4 ± 19.35 | 90.94 ± 19.68 |
| QT Corrected | 445.54 ± 35.44 | 445.31 ± 35.27 | 447.11 ± 36.6 |
| QT-QTC Ratio | 0.88 ± 0.1 | 0.88 ± 0.1 | 0.88 ± 0.11 |

Table 3: Subject demographics. Summary statistics for continuous values are denoted with mean ± std dev. TTE: Transthoracic echocardiogram. ECG: Electrocardiogram. CXR: Posteroanterior (PA) view chest X-ray. The "Other" subgroup comprises of subjects who identified as "American Indian or Alaska Nation", "Native Hawaiian Other pacific island", or "Other Combinations Not Described". SHD: Structural heart disease.

## Appendix B. Subgroup Performance Metrics

| Subgroup | Subjects (n) | SHD (%) | AUROC ↑ | AUPRC ↑ |
|---|---|---|---|---|
| Male | 708 | 19.49 | 0.802 (0.758,0.84) | 0.52 (0.428,0.599) |
| Female | 903 | 14.84 | 0.837 (0.799,0.874) | 0.521 (0.428,0.609) |
| White | 642 | 18.54 | 0.855 (0.813,0.893) | 0.628 (0.534,0.716) |
| Black / AA | 337 | 18.40 | 0.796 (0.736,0.85) | 0.47 (0.345,0.591) |
| Asian | 57 | 14.04 | 0.809 (0.627,0.96) | 0.604 (0.199,0.888) |
| Other | 325 | 11.38 | 0.786 (0.706,0.86) | 0.378 (0.23,0.529) |
| Declined | 147 | 15.65 | 0.861 (0.772,0.938) | 0.598 (0.37,0.793) |
| Missing | 103 | 22.33 | 0.724 (0.62,0.829) | 0.467 (0.287,0.647) |

Table 4: Performance metrics by subgroup for the classifier with ECG, CXR, and tabular inputs. Parentheses denote 95% confidence intervals. Black / AA: Black or African American.

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
