# OpenReview forum: "Does Multimodality Help in Deep Learning-Based Structural Heart Disease Detection?"
_MIDL.io/2024/Short_Papers — MIDL 2024 Short Papers_

### Official Review · Reviewer_Duij · 2024-04-24

**Confidence:** 5
**Final Rating:** 3.5

**Review:**

This study investigates what combination of common clinical modalities (i.e. ECGs, posteroanterior view chest X-rays, EHR) can detect structural heart disease labeled using transthorasic echocardiograms. The results show that the ECG-based models perform best and that the inclusion of other modalities can marginally improve the results.

The strengths of this work are:
1) the quality of the dataset used in this study
2) the relevance of the experiments and scores that have been chosen

The main weakness of this work concerns the absence of description and labeling of the different heart diseases present in the dataset. Indeed, the relevance of using a dedicated modality is strongly influence by the type of heart disease that is investigated. It would therefore have been preferable to calculate the different metrics for each pathology class. This could have an impact on the interpretation of the results.

---

### Decision · Program_Chairs · 2024-04-26

Accept